# Impact of Spatial Rainfall Scenarios on River Basin Runoff Simulation a Nan River Basin Study Using the Rainfall-Runoff-Inundation Model

Kwanchai Pakoksung 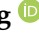

International Research Institute of Disaster Science, Tohoku University, Sendai 980-0845, Japan;
pakoksung@irides.tohoku.ac.jp; Tel.: +81-22-752-2090

**Abstract:** This study aims to investigate the impact of spatial rainfall distribution scenarios from ground observation stations on runoff simulation using hydrological modeling specific to the Rainfall-Runoff-Inundation (RRI) model. The RRI model was applied with six different spatial distribution scenarios of input rainfall, including Inverse Distance Weight (IDW), Thiessen polygon (TSP), Surface Polynomial (SPL), Simple kriging (SKG), and Ordinary kriging (OKG), to simulate the runoff of a 13,000 km$^2$ watershed, namely the Nan River Basin in Thailand. This study utilized data from the 2014 storm event, incorporating temporal information from 28 rainfall stations to estimate rainfall in the spatial distribution scenarios. The six statistics, Volume Bias, Peak Bias, Root Mean Square Error, Correlation, and Mean Bias, were used to determine the accuracy of the estimated rainfall and runoff. Overall, the Simple kriging (SKG) method outperformed the other scenarios based on the statistical values to validate with measured rainfall data. Similarly, SKG demonstrated the closest match between simulated and observed runoff, achieving the highest correlation (0.803), the lowest Root Mean Square Error (164.48 cms), and high Nash-Sutcliffe Efficiency coefficient (0.499) values. This research underscores the practical significance of spatial interpolation methods, such as SKG, in combination with digital elevation models (DEMs) and landuse/soil type datasets, in delivering reliable runoff simulations considering the RRI model on the river basin scale.

**Keywords:** rainfall spatial distribution; runoff simulation; Rainfall-Runoff-Inundation model

## 1. Introduction

The rainfall is significant data in hydrological processes and is well-established, particularly in the context of runoff modeling. Rainfall patterns, with regard to their spatial distribution and uniformity, are integral to shaping hydrological systems, particularly when utilizing distributed hydrological models. These models are designed to simulate and predict the behavior of water flow through intricate networks of interconnected channels, streams, and rivers. Therefore, accurate and reliable rainfall data are crucial to ensuring the effectiveness and accuracy of these models. Several studies have emphasized the importance of understanding the impact of rainfall characteristics on discharge estimation.

For instance, Goodrich et al. [1] highlighted the applicability of uniform rainfall in modeling runoff for small catchments, but they also stressed the significant influence of rainfall spatial distribution on discharge. Schuurmans and Bierkens [2] emphasized that relying on a single rainfall observation point outside the river basin watershed can lead to erroneous predictions and underscored the essential role of rainfall spatial data in runoff modeling. This paper aims to summarize the findings from eight different rainfall scenarios, investigating the effects of spatial distribution and uniformity on runoff simulation in a hydrological model for small catchments.

Precipitation characteristics have a considerable impact on the estimation of discharge. The spatial distribution of rainfall directly influences the accuracy of runoff prediction, particularly concerning the spatiotemporal resolution of rainfall [3]. Different precipitation

regimes, such as peak rainfall intensity and timing, can significantly impact peak discharge in catchments prone to debris flow [4]. Additionally, the temporal evolution of rainfall plays a crucial role in the hydrograph and pollutant discharge of combined sewer overflow (CSO) systems [5]. In storm drainages linked to sewage, the total rainfall and discharge volume are critical factors that affect the concentration and load of pollutants per unit area [6]. Furthermore, the variability of rainfall time distributions can impact peak discharge, with different rainfall time distribution patterns triggering unique response patterns in debris flows [7].

Moreover, Tsntikidis et al. [8] and Chintalapudin et al. [9] observed that several watersheds lack an adequate distribution of rainfall observation stations; it is challenging to consider the spatial variation of rainfall in the accuracy of runoff simulation. The relationship between precipitation and discharge, especially for frequent events, was emphasized by Arnaud et al. [10]. This study focuses on four artificial river basin watersheds ranging from 20 to 1500 km$^2$ and evaluates the data with three hydrological models. Drawing from the work of Bell and Moore [11], they place particular emphasis on high spatial resolution rainfall data, especially in the context of convective rainfall events.

The distribution of rainfall observation stations significantly influences the accuracy of runoff simulations. In particular, the spatial variability of precipitation input, especially for short-duration rainfall events and small catchments, substantially impacts modeling performance [12]. Increasing the station density to a specific resolution may improve modeling accuracy, depending on the catchment area, event type, and station distribution [13]. The number and location of rainfall stations play a vital role in modeling rainfall-runoff, with optimal results obtained using a sub-network of well-distributed stations [14]. Furthermore, characteristics of rainfall spatial distribution, such as normalized root-mean-square error, skewness, and second-scaled spatial moment, determine the lower limit of rainfall spatiotemporal resolution for runoff models and runoff prediction accuracy [15]. Incorporating spatial information about rainfall distribution enhances the performance of deep-learning models in runoff simulations [16].

The Rainfall Runoff Inundation (RRI) model is an essential hydrological tool for predicting flood events and assessing the potential inundation areas caused by heavy rainfall. Widely used in various regions, the model facilitates flood discharge simulation, flood return period prediction, and economic loss analysis caused by flooding. The RRI model integrates rainfall data from several sources, including ground stations and satellites, and topographic and land data from remote sensing. The model provides valuable information for flood mitigation and disaster management by incorporating rainfall-runoff and flood inundation simulations. As such, it is an indispensable tool for professionals in hydrology and disaster management.

Several studies have applied the RRI model to simulate hydrological processes and generate runoff. Notably, Sayama et al. [17] established and verified the RRI model's effectiveness by comparing simulated inundation maps with MODIS satellite data for flood events in large-scale river basins. Applications of the RRI model for runoff simulation in the Chao Phraya River Basin, Thailand, were mentioned by Shakti et al. [18]; also, the RRI was demonstrated by including the 2011 flood event, which was used to estimate damage costs [19].

The RRI model has been effectively utilized in the Batang Sinamar River Basin located in Lima Puluh Kota Regency [20], as well as in Jakarta and the Ciliwung-Cisadane Watershed [21], and the Bekasi River Basin situated in Bekasi City [22]. Additionally, a budget-friendly deep learning model called Rain2Depth has been created to mimic the RRI model and estimate the maximum inundation depth's spatial distribution [23]. The model has been applied to simulate flooding and the impact of land cover changes on inundation in the Indus River basin [24].

In this context, the primary point of this study is to examine how spatial rainfall scenarios impact runoff simulation using the Rainfall-Runoff-Inundation (RRI) model. The study area is a catchment covering 13,000 km$^2$ in northern Thailand and utilizes rainfall

data from 2014 to interpolate spatial and temporal rainfall patterns. This analysis spans wet and dry seasons, offering insights into river basin catchment behavior. We conducted a sensitivity study, comparing different rainfall datasets as the input data for the RRI model and evaluating the simulated runoff with observed discharge hydrographs. It is significant to reveal that this study points to sensitivity analysis without calibration and assesses the effect of varying rainfall data inputs on the RRI model's performance.

This introduction provides an overview of the paper's context, objectives, and key findings. Section 2 delves into the characteristics of the study area and the hydrological model used. Section 3 details the interpolation methods, performance metrics, and model setup. Section 4 presents this study's results, featuring discharge hydrographs, and Section 5 offers conclusions and a discussion of the findings.

This study highlights the effectiveness of hydrological modeling in identifying the sensitivity of spatial rainfall to river basin responses. However, the accuracy of such an analysis depends on the precise replication of the watershed response by the employed model. This research offers valuable insights into daily rainfall variability in spatial terms, aiming to improve runoff predictions in hydrological modeling at the scale of river basin watersheds.

## 2. Data Sets and Methods

### 2.1. Nan River Basin

The upstream area of the Nan River Basin is considered by the study area to be a region of great significance due to its role in supplying water from the SIRIKIT dam to Thailand's central area, including the capital area, Bangkok City. Figure 1 describes the study area in northern Thailand, with a total watershed area of approximately 13,000 km$^2$. The main river originates in the Bor-Klua District, Nan Province, at the north of the area, between latitude 17d 42′12″ N and latitude 19d 37′48″ N and longitude 100d 06′30″ E to longitude 101d 21′48″ E. Approximately 88% of the region consists of mountainous terrain, while the remaining 12% is inhabited by residents within the watershed. Downstream from the SIRIKIT dam, which serves as the modeling river outlet, the river bed features a steep slope of approximately 1/1500. Moving upstream, the slope becomes flat (1/10,000), followed by a steep slope (1/600). The elevation in this area ranges from 70 to 1200 m above mean sea level and has a mean annual rainfall of approximately 1380 mm. The Wa River, Nam Pua River, and Nam Yao River are essential tributaries contributing to the complex hydrological system. Fluvial has been observed in the Tawang Pha, Muang Nan, and Wiang Sa areas as vulnerable areas.

### 2.2. Rainfall-Runoff-Inundation Model

The model used to simulate runoff in this study is the Rainfall-Runoff-Inundation (RRI) model, a two-dimensional model that models surface and river channels separately [17]. The model sets the river channel within a grid pixel to consider the surface and river at the same grid pixel. The river channel was set by a line along the centerline of the pixel of the surface grid. The represented channel provides an additional flow path between grid pixels along the river line. The simulation of lateral flows is estimated on the surface grid by the two-dimension approach. The river channel in each grid contains two water depths: channel and surface. In the channel grid, inflow and outflow were estimated by the overflow condition, water level, and levee height.

To solve the two-dimensional scheme of the RRI model, it applied diffusive wave routing using the numerical method of the fifth order of the Runge-Kutta algorithm. This model estimates water depth on the surface pixel based on a difference in water level between the upstream grid and downstream grid, where the water level is the combination of water depth and ground surface level. The model requires three main inputs: rainfall data in temporal grid format, topography data (Digital Elevation Model (DEM), Flow direction, and Flow accumulation), and landuse/soil type data for infiltration processes.

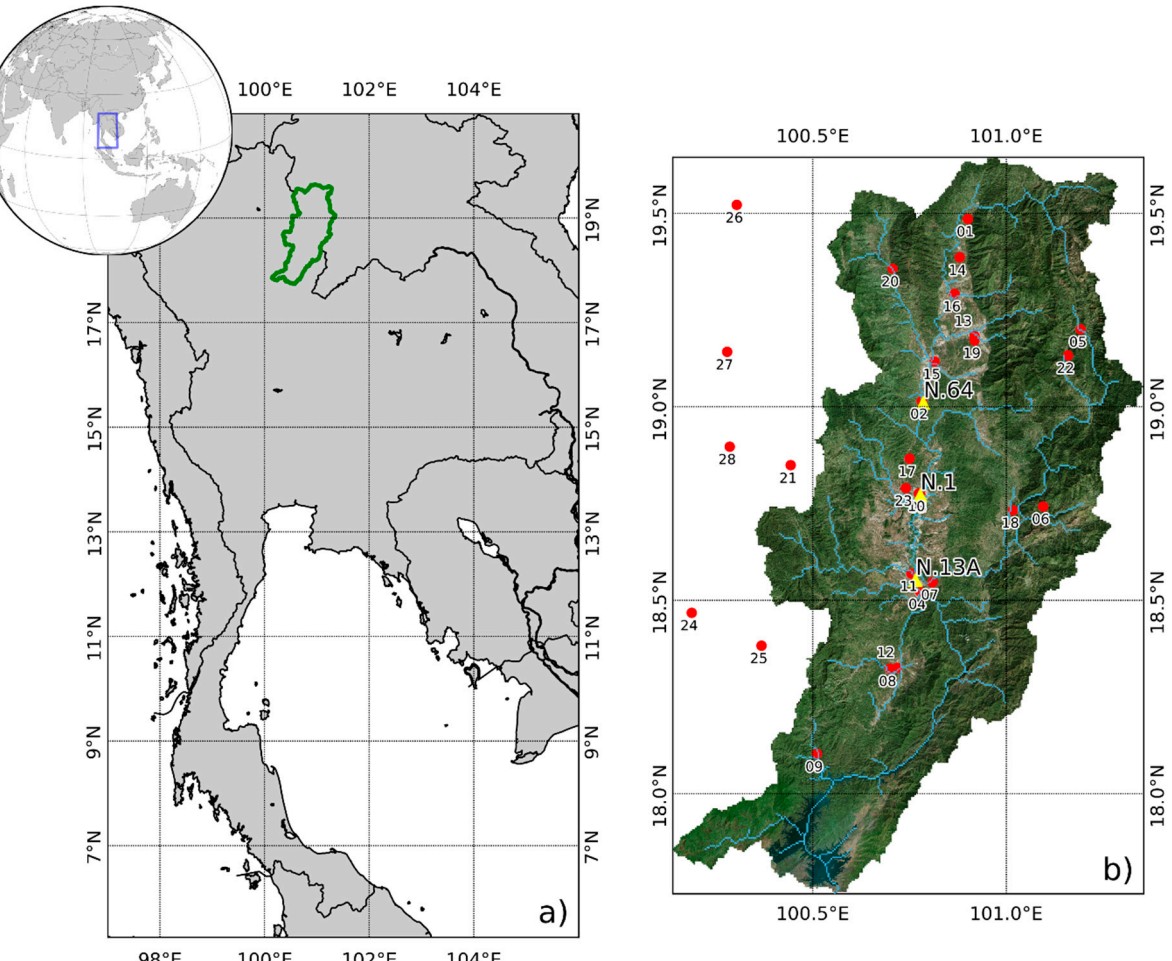

**Figure 1.** Location of this study area. (**a**) The Nan River Basin location in Thailand. (**b**) The Nan River Basin in detail: the red dot represents the observation station for rainfall, and the yellow triangle represents the runoff observation station.

*2.3. Storm Events and Rainfall Observation Data*

Over the last 20–30 years, Northern Thailand has seen an increasing trend in precipitation levels [25]. The period of the year characterized by rainfall, typically spanning from June to September, has been observed to commence earlier and extend further, encompassing May to October [26]. In the region, the mean annual rainfall values have been discovered to be most pronounced in the northeastern sector, gradually diminishing towards the southwest [27]. Furthermore, applying clustering techniques to assess the precipitation patterns within this area has also revealed the presence of the highest rainfall zone in the northern section, displaying moderate precipitation levels in the southeastern region and the lowest levels in the southwestern area [28]. Consequently, these findings suggest an overall escalation in precipitation in Northern Thailand over the preceding few decades, alongside variations in rainfall distribution throughout different sections of the region [29].

This study focuses on the 2014 events, which were instrumental in evaluating different rainfall products used in the RRI model for the basin. These events included tropical storms, thunderstorms, and significant rainfall associated with weather patterns from the Pacific Ocean that traveled westward to the area from March to August. The Inter-Tropical Convergence Zone (ITCZ) typically impacts the northern area of Thailand from May to August, as mentioned by Schneider et al. [30]. Conspicuously, the monsoon event from 28 to 30 August 2014 led to heavy rainfall, ranging from 100 to 150 mm, causing severe flooding and river bank overflow.

Rainfall observations were collected from 28 stations, as indicated in Figure 1b, with 17 stations located within the watershed and 11 on the western side. These stations provided daily temporal data that drove the RRI model throughout 2014. The spatial distribution of rainfall was constructed using these 28 stations, as detailed in Section 3.

### 2.4. Topography Data

Topography data were provided by the United States Geological Survey (USGS) and sourced from the Shuttle Radar Topography Mission (SRTM). The SRTM project was a collaboration between the National Imagery and Mapping Agency (NIMA) and the National Aeronautics and Space Administration (NASA). The SRTM data are online from the Consultative Group for International Agriculture Research Consortium for Spatial Information (CGIAR-CSI). The provided DEM data covers approximately 80% of the earth's surface, from a latitude of 60 degrees in the north to 60 degrees in the south. This DEM provides a one-arc-second resolution of approximately 30 m to contain 16 m in vertical accuracy at 90% confidence and 20 m in horizontal accuracy at 90% confidence, as reported by Javis et al. [31]. In this study, the original SRTM was upscaled to 500 m pixel resolution (approximately 15 × 15 arc-seconds). This pixel size encompasses 457 rows and 292 columns, representing the watershed area of 13,000 km$^2$, as shown in Figure 2a. The specific SRTM data for the study area was obtained under index No. srtm_57_09.

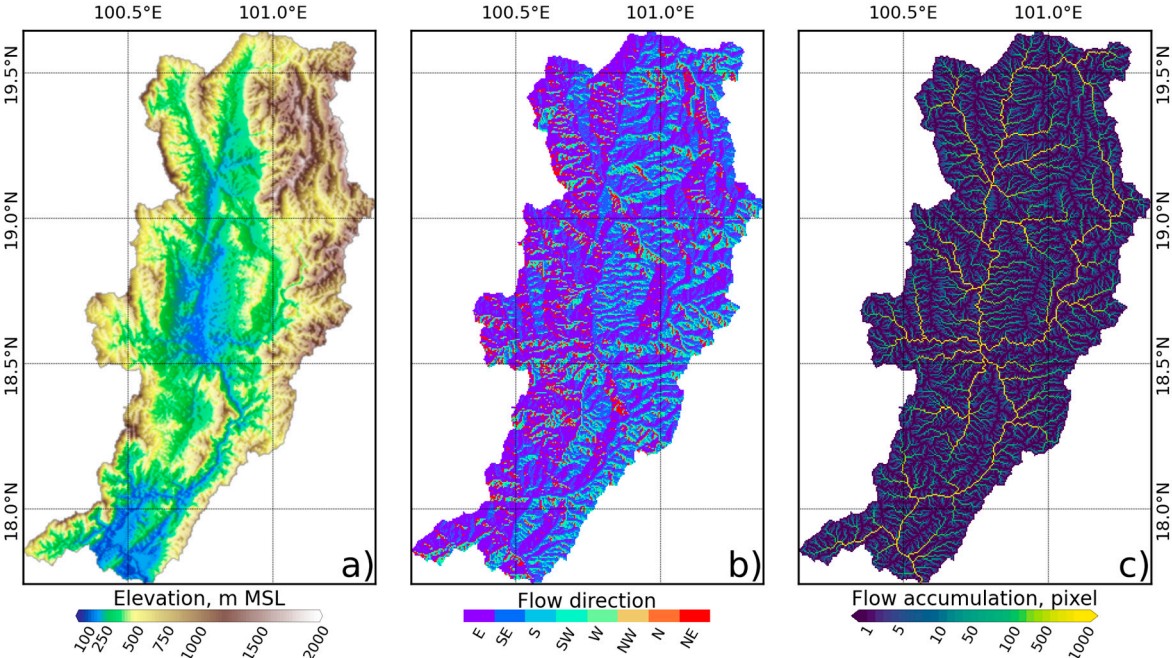

**Figure 2.** Topographical characteristics of the Nan River Basin. (**a**) Topography; (**b**) Flow direction; and (**c**) Flow accumulation.

Flow direction data were derived from the slope of the DEM to establish downstream direction based on the eight surrounding directions, as shown in Figure 2b. The flow accumulation data, presented in Figure 2c, was calculated from the flow direction data, representing the number of upstream areas for a given point in the form of grid cells. These values are crucial for understanding the channel network within the basin.

### 2.5. Landuse and Soil Types

The USGS provided landuse data through the Global Land Cover Characteristics (GLCC), and soil type data were provided by the Land Development Department (LDD) in Thailand. Both datasets were resampled to match the resolution of the DEM through projection. The GLCC dataset was observed in 2000 to comprise the NDVI composite of

1 km of AVHRR data. This data are essential for landuse characterization, covering the period from April 1992 to March 1993 [32]. Figure 3a shows that the landuse categories were classified into six types based on the GLCC data. Forested areas dominate the study area, accounting for 70.69%, and are predominantly located along the watershed's border. Approximately 14.58% of the land is cropland, followed by grassland at 9.61%, water bodies at 2.1%, deforested areas at 1.91%, and urban areas at 1.1%. The cropland and urban areas are primarily distributed along the floodplain area in the upstream and middle paths of the river basin.

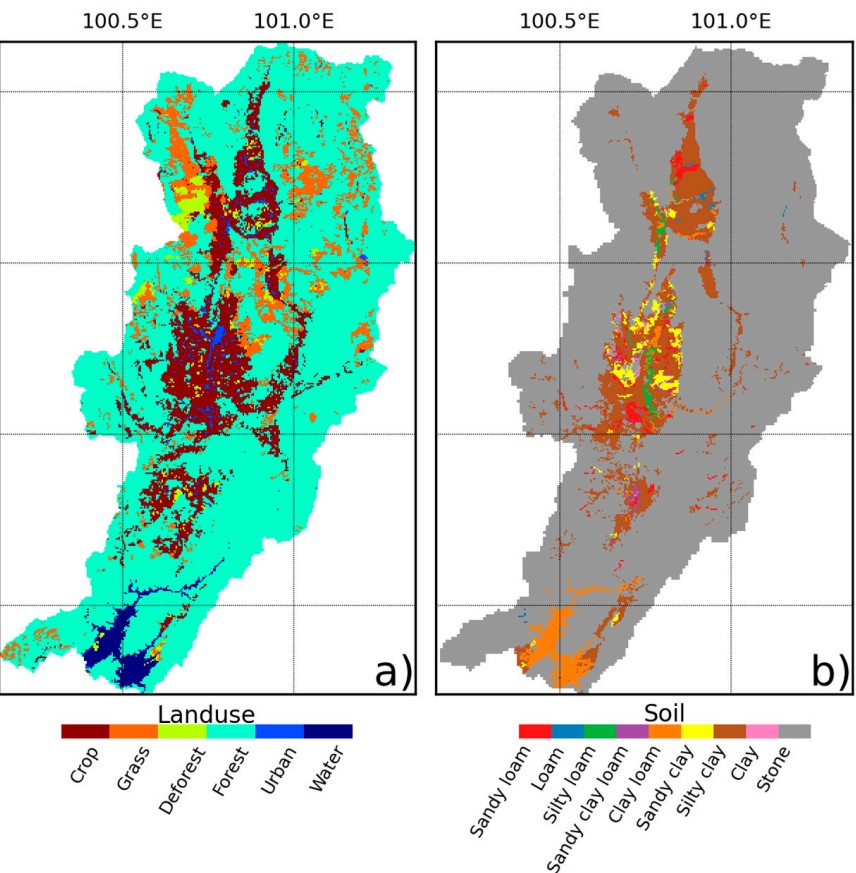

**Figure 3.** (**a**) Landuse and (**b**) Soil type in the Nan River Basin, Thailand.

Soil type data were provided by the LDD [33], Thailand, and categorized into nine types. Most of the soil is represented by the stone type, which is located in mountainous areas and constitutes 83.68% of the total. The floodplain areas have eight different soil types: silty clay, clay loam, sand clay, sandy loam, silty loam, sand clay loam, loam, and clay. This is shown in Figure 3b.

## 3. Methodology

The methodology, illustrated in Figure 4, aims to address the primary research question of this study. We utilized daily rainfall spatial distribution data from five different scenarios in 2014 to answer this question. These scenarios encompassed three deterministic spatial methods and two geostatistical spatial methods. The performance of these methods was assessed by comparing their outputs to observed discharge data at the runoff station (Figure 1b) using five statistical coefficients.

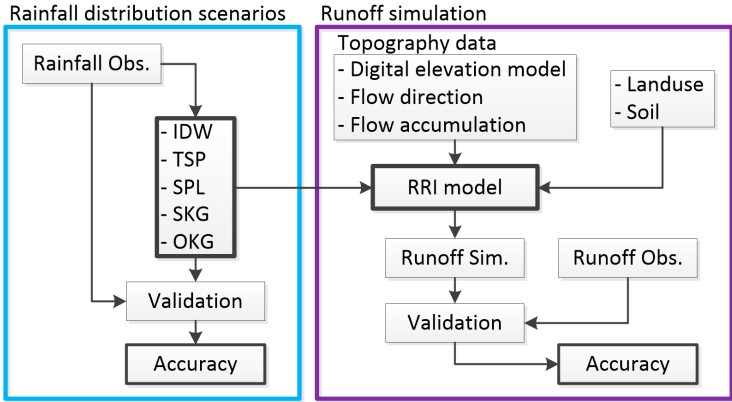

**Figure 4.** Methodology streamlines of this study.

*3.1. Rainfall Spatial Scenario*

Two methods of interpolation were employed: deterministic and geostatistical methods. The deterministic method was selected by three algorithms: Inverse Distance Weighting, Thiessen Polygon, and Surface Polynomial. The geostatistical method was selected by two algorithms: Simple Kriging and Ordinary Kriging. All algorithms were implemented in Python programming to reproduce the daily rainfall distribution in a grid format for the 2014 event. Five statistical indexes evaluated the performance of the considered algorithm.

Typically, spatial interpolation involves estimating values at sampling points based on the weighted values of surrounding observed points. The sampling points are located at the centers of the regular grid with a resolution of 0.1 × 0.1 degrees, covering the study area and featuring 13 rows and 20 columns. The spatial interpolation formula takes the following forms for each method:

$$R_e = \sum_{i=1}^{n} w_i \cdot R_i, \tag{1}$$

where $R_e$: the interpolated value at point $e$; $R_i$: the measured value at point $i$; $n$: the number of measured points as rainfall stations in the study area; $w$: the weight of rainfall station for interpolation. The coordinates of the rainfall stations are $x_i$ and $y_i$ in two-dimensional, and $R_i$ is related on $x_i$ and $y_i$. Equation (1) can be applied to interpolate $R_e$ in any coordinate $x_e$ and $y_e$ [34].

3.1.1. Inverse Distance Weight (IDW)

The IDW algorithm interpolates values at a sampling point using the average weight of the measured points from the neighboring points. The weight is inversely proportional to distance, increasing as the distance to the observed point decreases [35]. The weight can be estimated by:

$$w_i = \frac{\frac{1}{D_i^k}}{\sum_{i=1}^{n} \frac{1}{D_i^k}}, \tag{2}$$

where $D_i$: the distance between the measured and considering point, and $k$: friction distance is assumed as 2 in this study [36]. The Inverse Distance Square method is revealed by the friction distance value of 2.

3.1.2. Thiessen Polygon (TSP)

The TSP algorithm is considered by the nearest neighbor method [37] and estimates values at unknown points (the centers of rainfall grids) by using the value of the nearest observed point. The weight is determined based on the closest distance. The weight can be estimated by:

$$D_{ei} = \sqrt{(x_e - x_i)^2 + (y_e - y_i)^2}, \tag{3}$$

for $D_j = min(D_{e1}, \ldots, D_{en})$ and then $w_i = 0$ for $i \neq j$ while $w_i = 1$ for $i = j$. This algorithm is revealed to be a simple algorithm and is unsuitable for mountain areas due to the orographic effect of the rain [38].

### 3.1.3. Surface Polynomial (SPL)

The SPL algorithm involves fitting a global function by using an algebraic and trigonometric polynomial method. In this study, we used a cubic spline method with a degree of three. The general function is followed by:

$$R_e(x, y) = \sum_{k1=0}^{m} \sum_{k2=0}^{m} a_{k1,k2} \cdot x^{k1} \cdot y^{k2}, \tag{4}$$

where $a_{k1,k2}$: the $k1$ and $k2$ is polynomial coefficient; $x$ and $y$ are the coordinates of the estimated point; and $m$: number of polynomial functions to fit degree. From the study of Tabios and Salas [39], the weight value of each measured point is calculated using the least squares method. The equation form is followed by:

$$w_i = \sum_{k1=0}^{m} \sum_{k2=0}^{m} a_{k1,k2,i} \cdot x^{k1} \cdot y^{k2}, \tag{5}$$

The polynomial function's coefficients can be estimated by the inverse matrix algorithm. The number of polynomial functions fitted in degrees is used as 3 in this study, as considered by the Cubic Spline method.

### 3.1.4. Simple Kriging (SKG)

The SKG algorithm is a method of using the mean sampling data set and the semi-variogram model to estimate the weight at each measured point. The weight equation is followed by:

$$w_i = \gamma_i \left[ C - \gamma(h_{ij}) \right]^{-1} \cdot (C - \gamma(h_{ip})), \tag{6}$$

where $\gamma(h_{ij})$: the semi-variogram model from point $i$ to $j$; $h_{ip}$: the semi-variogram distance of measured and interpolated point.

### 3.1.5. Ordinary Kriging (OKG)

The OKG algorithm is a linear geostatistical method that calculates weights based on minimized variance and provides an unbiased value. The weight is followed by:

$$w_i = \gamma(h_{ij})^{-1} \cdot \gamma(h_{ip}); \sum_{i=1}^{n} w_i = 1, \tag{7}$$

where $\gamma(h_{ij})$: the semi-variogram from point $i$ to $j$; $h_{ip}$: the semi-variogram distance of measured and interpolated point. Based on the unbiased estimation value, the constraint function of the weight is equal to 1.

### 3.1.6. Semi-Variogram Model

The semi-variogram model is fundamental in geostatistical methods for representing spatial correlations at a spatial point [40–44]. This model is applied to analyze the variance in distance between all pairs of sampling points. The semi-variogram equation is followed by:

$$\hat{\gamma}(h) = \frac{1}{2N(h)} \sum_{i=1}^{N(h)} (R_i - R(U_i + h))^2, \tag{8}$$

where $N(h)$: the number of pairs separated by lag $h$; $U$: vector of spatial coordinates. The semi-variogram model in this study was estimated by 28 rainfall stations with a daily rainfall temporal scale. The estimated semi-variogram on a pooled semi-variogram was

fitted using a spherical semi-variogram model. The semi-variogram model function is followed by:

$$\gamma(h) = \begin{cases} C_0 + C\left(\dfrac{3h}{2a} - \dfrac{1}{2}\left(\dfrac{h}{a}\right)^3\right), & 0 < h \leq a \\ C_0 + C, & h > a \\ 0, & h = 0 \end{cases},$$
(9)

The spherical variogram model parameters are followed by the nugget variance ($C_0$) 0.425, the partial sill ($C$) 1.404, and the range ($a$) 0.545. The model's coefficients were used to estimate the weight through Equation (1) with two different geostatistical methods: simple kriging and ordinary kriging.

### 3.2. Simulation Model Setup

The RRI model's input data comprises three data types: rainfall, topography, and landuse/soil type. The hydrological parameters were recommended from previous studies [45], which are based on calibration in previous RRI modeling studies [46]. Table 1 shows Manning's roughness coefficient related to the landuse type for the flow routing processes in the RRI model. Table 2 presents the Green-Amp parameter related to the soil type for the infiltration processes in the RRI model. These parameters were applied to simulate runoff through the RRI model in the rainfall distribution scenarios.

**Table 1.** Landuse type is related to the n's Manning coefficient.

| Landuse Type | n's Manning |
|---|---|
| Forest | 0.50 |
| Deforestation | 0.40 |
| Grasslands | 0.30 |
| Cropland | 0.35 |
| Urban and Build-up | 0.05 |
| Water bodies | 0.04 |

**Table 2.** Soil type is related to soil parameters for the RRI model.

| Soil Type | Soil Depth, m | Saturated Hydraulic Conductivity (ka), cm/h | Beta, ka/kc | Green-Ampt Parameter | | |
|---|---|---|---|---|---|---|
| | | | | Ksv, cm/h | Porosity | Capillary Head, cm |
| Clay | 1.0 | 0.462 | | 0.06 | 0.475 | 31.63 |
| Clay loam | 1.0 | 0.882 | | 0.20 | 0.464 | 20.88 |
| Loam | 1.0 | 2.500 | | 1.32 | 0.463 | 8.89 |
| Sandy clay | 2.0 | 0.781 | | 0.12 | 0.430 | 23.90 |
| Sandy clay loam | 1.5 | 2.272 | | 0.30 | 0.398 | 21.85 |
| Sandy loam | 1.5 | 12.443 | | 2.18 | 0.453 | 11.01 |
| Silty clay | 1.0 | 0.366 | | 0.10 | 0.430 | 29.22 |
| Silty loam | 1.0 | 2.591 | | 0.68 | 0.501 | 16.68 |
| Stone | 1.5 | - | | - | - | - |

The characteristics of the river channel, width, and depth are estimated by the flow accumulation represented by the upstream area of the consideration point. The flow accumulation is based on the flow direction created from the resampled DEM. The flow direction is represented by eight directions: E, SE, S, SW, W, NW, N, and NE, around the consideration point [47,48]. The width and depth of the river channel are based on the following:

$$W = 16.93 \cdot A_{basin}^{0.186},$$
(10)

$$D = 16.93 \cdot A_{basin}^{0.120},$$
(11)

where $W$: the width of river channel in m, $A_{basin}$: the upstream area of the river channel point in km$^2$, and $D$: the depth of river channel in m.

### 3.3. Performance Statistics

The performance of the spatial distribution scenarios applied to the daily rainfall data were evaluated by six key performance statistics: Volume Bias, Peak Bias, Root Mean Square Error, Correlation Coefficient, Mean Error, and Nash-Sutcliffe Efficient coefficient. These five statistics were applied to estimate the accuracy of interpolated rainfall and simulated runoff [49].

Volume Bias ($V_{bias}$) and Peak Bias ($P_{bias}$) measure the systematic bias of modeled rainfall and simulated runoff in percentage (%).

$$V_{bias} = \frac{|Q_{vo} - Q_{vs}|}{Q_{vo}} \times 100, \tag{12}$$

$$P_{bias} = \frac{|Q_{po} - Q_{ps}|}{Q_{po}} \times 100 \tag{13}$$

Correlation Coefficient ($r$) quantifies the degree of correlation between two datasets, with 0 indicating no correlation and 1 representing a perfect correlation.

$$r = \frac{\sum_{i=1}^{n} \left( (Q_{oi} - \overline{Q_o}) \cdot (Q_{si} - \overline{Q_s}) \right)}{\sqrt{\sum_{i=1}^{n} (Q_{oi} - \overline{Q_o})^2 \cdot \sum_{i=1}^{n} (Q_{si} - \overline{Q_s})^2}}, \tag{14}$$

Root Mean Square Error ($RMSE$) measures the magnitude of differences between two datasets.

$$RMSE = \sqrt{\frac{\sum_{i=1}^{n} (Q_{oi} - Q_{si})^2}{n}}, \tag{15}$$

Mean Error ($ME$) quantifies the bias between two datasets.

$$ME = \frac{\sum_{i=1}^{n} (Q_{oi} - Q_{si})}{n}, \tag{16}$$

Nash-Sutcliffe Efficiency coefficient ($NSE$) is used to evaluate model efficiency by comparing the variance of the simulated time series to the variance of the observed time series, with 1 representing a perfect fit.

$$NSE = 1 - \frac{\sum_{i=1}^{n} (Q_{oi} - Q_{si})^2}{\sum_{i=1}^{n} (Q_{oi} - \overline{Q_{oi}})^2}, \tag{17}$$

where $Q_{vo}$: total volume of measured data, $Q_{vs}$: total volume of simulated data, $Q_{po}$: peak of measured data, $Q_{ps}$: peak of simulated data, $Q_o$: measured data, $Q_s$: simulated data, and $n$: number of sampling data.

## 4. Results and Discussion

### 4.1. Rainfall Spatial Distribution Interpolation

Rainfall distribution scenarios were estimated using five spatial interpolation methods: Inverse Distance Weighting (IDW), Thiessen Polygon (TSP), Surface Polynomial (SPL), Simple Kriging (SKG), and Ordinary Kriging (OKG). These datasets contained the same spatial and temporal resolution, with a grid size of 0.01 degrees (equivalent to 10 km) and daily time steps (24 h). Figure 5 illustrates a comparison of the rainfall at each observation for the year 2014. All scenarios exhibited patterns generally similar to the observed data, albeit with some variability. Conspicuously, OKG was slightly overestimated, while TSP was underestimated. Furthermore, all scenarios underestimated the peak.

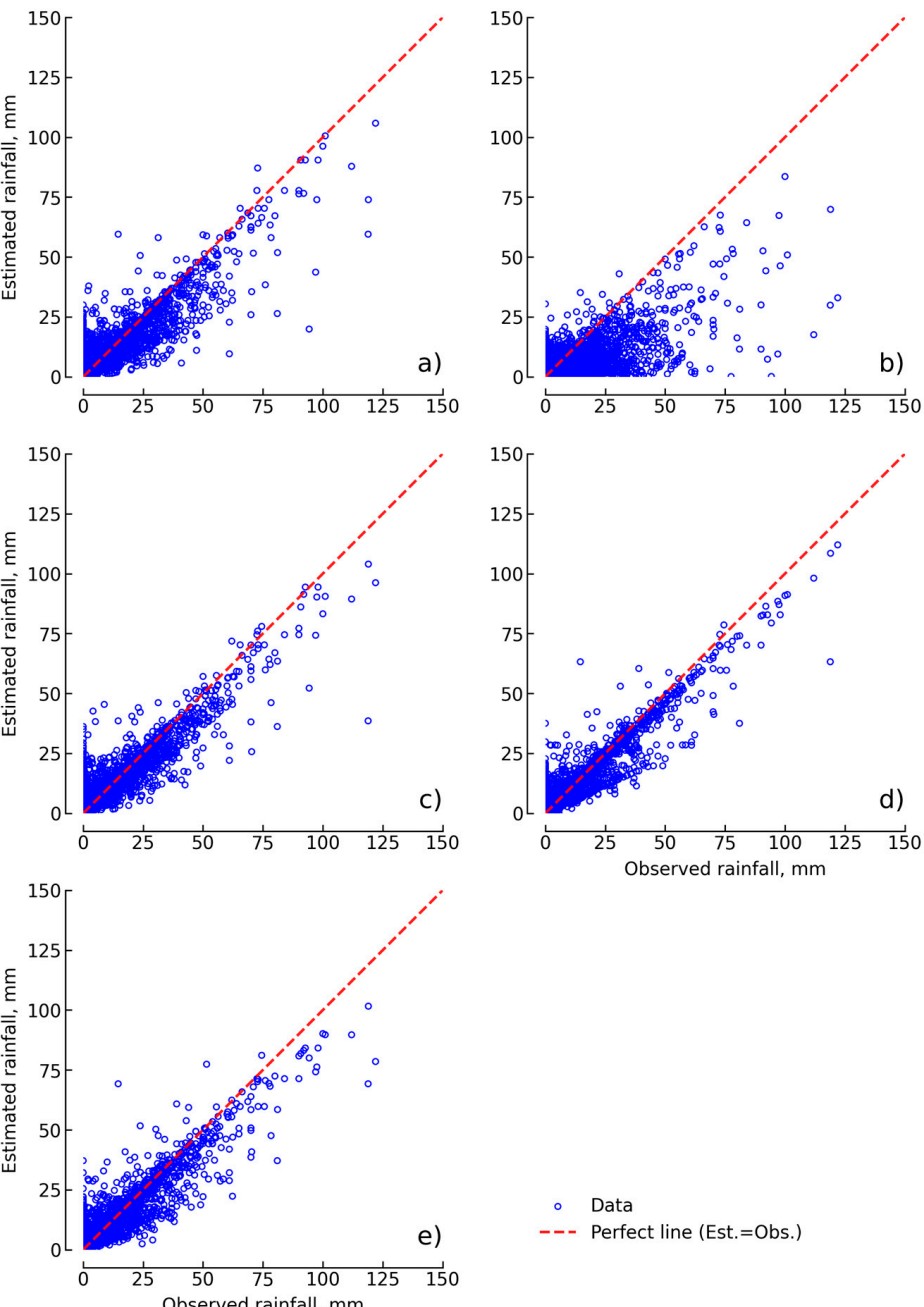

**Figure 5.** Scatter plot of rainfall data for observed and estimated distribution scenarios: (**a**) IDW, (**b**) TSP, (**c**) SPL, (**d**) SKG, and (**e**) OKG.

Figure 6 shows the spatial distribution of rainfall for all five scenarios. In general, all scenarios displayed spatial patterns that were similar to the actual event. In specific details, IDW, TSP, OKG, and SPL exhibited their maximum rainfall intensity in the northeastern part, along the watershed's border (both inside and outside), while SKGs maximum intensity was located within the watershed's northern part. The discrepancies in SKGs spatial interpolation were attributed to variations in the number and distribution of measured rainfall data points, affecting the quality of the semi-variogram model.

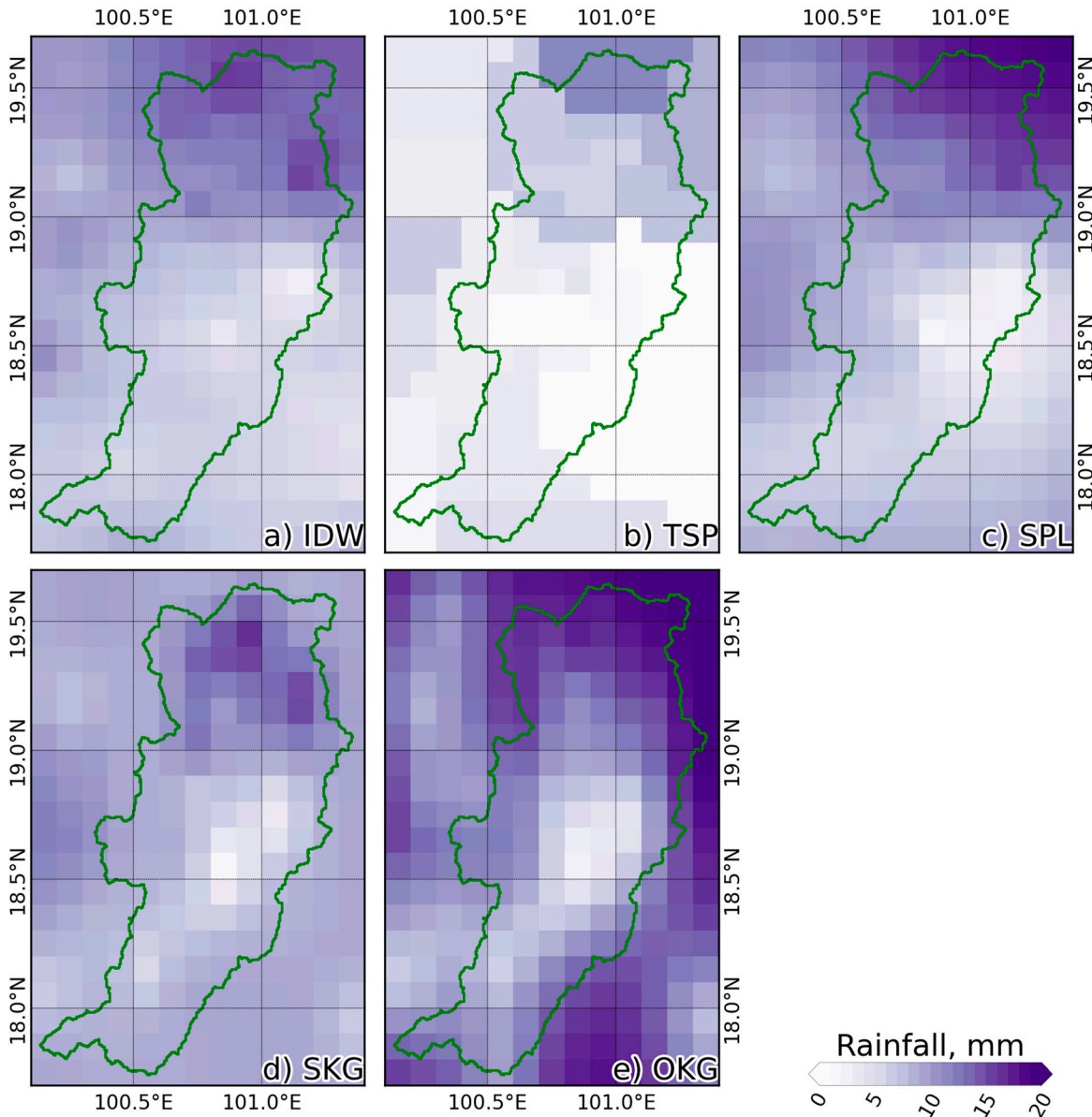

**Figure 6.** Average rainfall spatially for each distribution scenario.

Table 3 presents the total volume of rainfall for the five scenarios, calculated over the watershed area. The total volume of the observed rain gauge was approximately 7900.01 million cubic meters (MCM). OKG produced the highest value, followed by SKG, while TSP yielded the lowest. The maximum rainfall volume recorded during this event using OKG was approximately 10,654 MCM.

**Table 3.** Rainfall Volume for five distribution scenarios over the watershed area.

| Rainfall Distribution Scenario | Rainfall Volume, MCM |
|---|---|
| OBS | 7900.01 |
| IDW | 7045.41 |
| TSP | 3012.01 |
| SPL | 7578.28 |
| SKG | 7787.04 |
| OKG | 10,653.01 |

The accuracy of all interpolation scenarios was moderate, as evidenced by significant differences between observed and interpolated data, as summarized in Table 4. SKG is exhibited by the best correlation coefficient of approximately 0.95, followed by OKG, SPL, IDW, and TSP. According to RMSE, the SKG scenario outperformed the other. This superior performance of kriging methods was consistent with previous research. Mainly, OKG was the best at capturing peak values among the interpolation scenarios. All interpolation scenarios underestimated observed daily rainfall, as indicated by the negative values of Volume and Mean Bias. IDW, TSP, and SKG underestimated the measured data, while the two scenarios (SPL and OKG) overestimated it. Among the scenarios, TSP exhibited the highest negative value of Volume Bias and Mean Bias in underestimating observed rainfall by 59.47%. For NSE consideration, all estimated rainfall scenarios had a negative value close to zero, which reveals a significant difference between the estimated error variance and the observation variance. The best NSE value of approximately $-0.049$ is the SKG, followed by the OKG of approximately $-0.048$. The study area is a complex topography featuring elevations ranging from 100 to 2000 m above sea level over a short distance of approximately 200 km, highlighting TSPs limited suitability. This factor contributed to TSPs notable underestimation due to orographic effects on rainfall [50].

**Table 4.** Performance statistics for each rainfall distribution scenario on the rainfall estimation.

| Distribution Scenario | Volume Bias, % | Peak Bias, % | RMSE, mm | Correlation | Mean Bias, mm | Nash-Sutcliffe |
|---|---|---|---|---|---|---|
| IDW | −1.84 | −18.35 | 7.66 | 0.85 | −0.24 | −0.053 |
| TSP | −59.47 | −60.23 | 11.22 | 0.80 | −5.51 | −0.073 |
| SPL | 0.38 | −23.66 | 7.11 | 0.88 | 0.05 | −0.047 |
| **SKG** | **−0.18** | −20.12 | **5.08** | **0.92** | **−0.02** | **−0.049** |
| OKG | 4.39 | **−12.94** | 6.86 | 0.90 | 0.46 | −0.048 |

Bold is the highlight of the highest accuracy of the distribution scenario.

The accuracy of the five scenarios over the Nan River Basin, Thailand, particularly the spatial distribution, is assessed using RMSE as presented in Figure 7. Figure 7 shows that SKG agreed best with the rain gauge data. It is interesting to reveal that the interpolation scenario performed best in the northern part of the watershed boundary, despite its high mountainous terrain, while the middle and southern areas had the lowest RMSE values for the five scenarios. SKGs superior performance in mountainous areas might be attributed to its semi-variogram model, which effectively accounted for the complex terrain and orographic rainfall [50].

Among all the spatial interpolation rainfall scenarios, the kriging method with the semi-variogram model offered the best agreement with observed rainfall in this study and was consistent with findings from other studies: Tabios and Salas [37], Ly et al. [39], and Ly et al. [44].

## 4.2. Runoff Data Resulted from Rainfall Spatial Distribution Interpolation

The RRI model represents the hydrological modeling that was applied to simulate the runoff from the selected storm events using a consistent set of hydrologic parameters. The five rainfall distribution scenarios were used to estimate daily runoff, matching the measured runoff data provided by the Royal Irrigation Department, Thailand. The runoff stations were limited by three measured stations along the Nan River: N.64 in the upstream area of the watershed, N.1 in the middle area, and N.13 in the downstream area, as shown in Figure 1b. Figure 8 presents the simulated daily runoff hydrograph for all five rainfall interpolation scenarios at the measured runoff stations. Among all scenarios during the dry season from January to April, the TSP revealed the simulated value close to the observation data, while other scenarios revealed overestimation. Using two sets of parameters for rainfall estimation specific to dry and wet seasons is recommended.

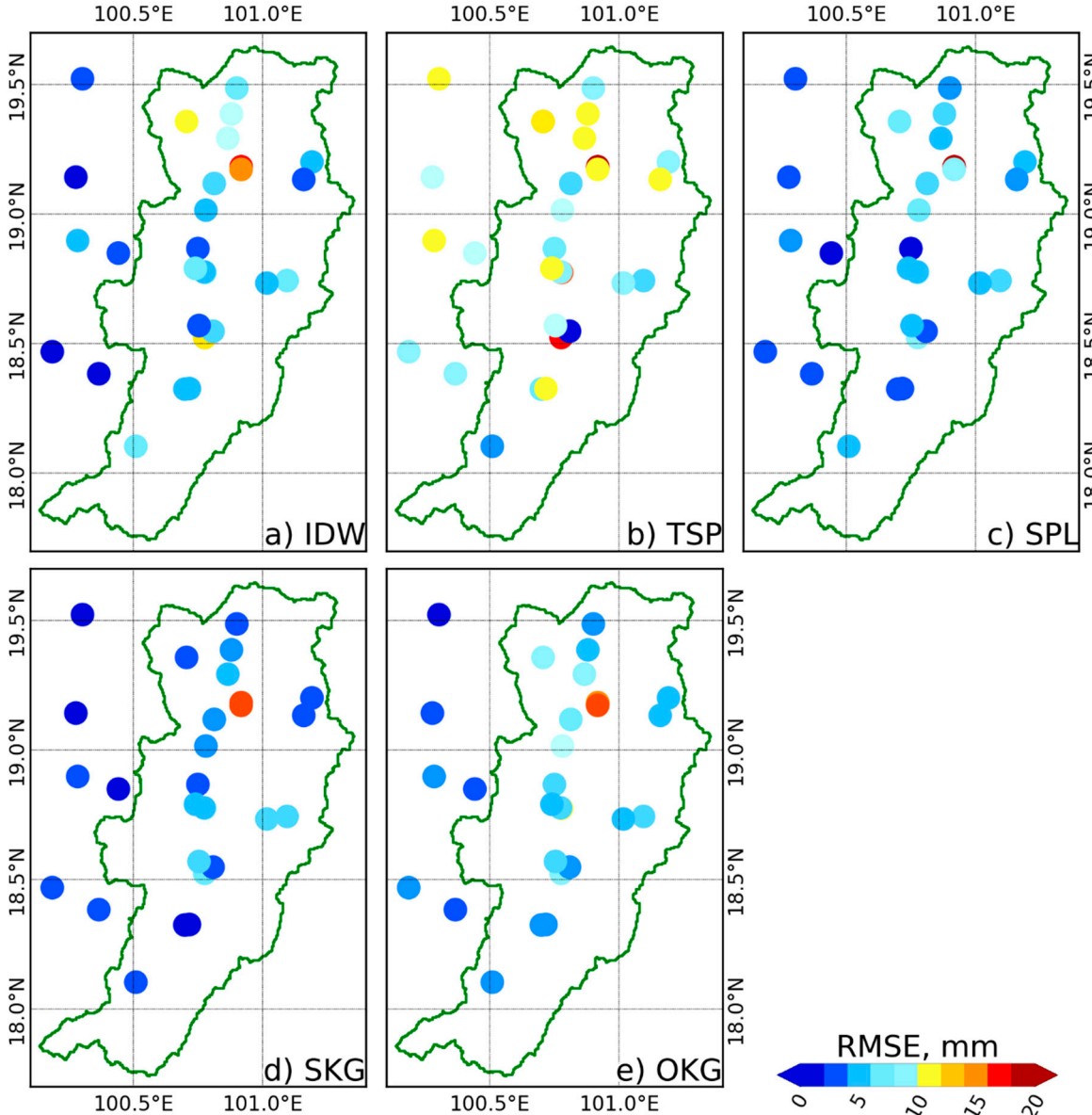

**Figure 7.** Root Mean Square Error (RMSE) at observed points for each rainfall distribution scenario on rainfall estimation.

All modeled runoff, generated using various interpolation scenarios, exhibited temporal patterns that closely resembled the observed hydrographs. However, the simulated runoff before the peak was consistently underestimated, while the simulated runoff after the peak was overestimated. OKG exhibited a significant overestimation of the peak and runoff volume for N.64 and N.1, although N.13 showed a slight underestimation. In contrast, TSP significantly underestimated runoff for all three stations.

All three runoff stations were assessed using performance statistics, as summarized in Table 5, which includes five performance indexes. SKG performed the best with measured runoff data, with the highest correlation of approximately 0.803 and the lowest RMSE of approximately 164.48 cm. Nevertheless, SKG overestimated runoff volume and mean runoff while underestimating peak flow by 6.8%, 19.4 cubic meters per second, and −35.8%, respectively. OKG yielded the lowest peak flow, but its simulated runoff volume exhibited the highest bias. SPL reveals an overestimation of runoff volume and mean runoff, considering Volume Bias and Mean Bias, respectively. In contrast, the SPL reveals an underestimated peak flow, resulting in a high RMSE with a strong correlation. TSP exhibited

a high RMSE value and moderate correlation, while runoff volume, peak flow, and mean runoff are underestimations. IDW overestimated runoff volume and mean runoff while underestimating peak flow, with a low RMSE and the lowest correlation value. To consider in the NSE evaluation, all scenarios reproduced the NSE value less than 0.5, and only OKG was the negative value. The best NSE value among all scenarios was SKG, approximately 0.499, followed by IDW, approximately 0.437.

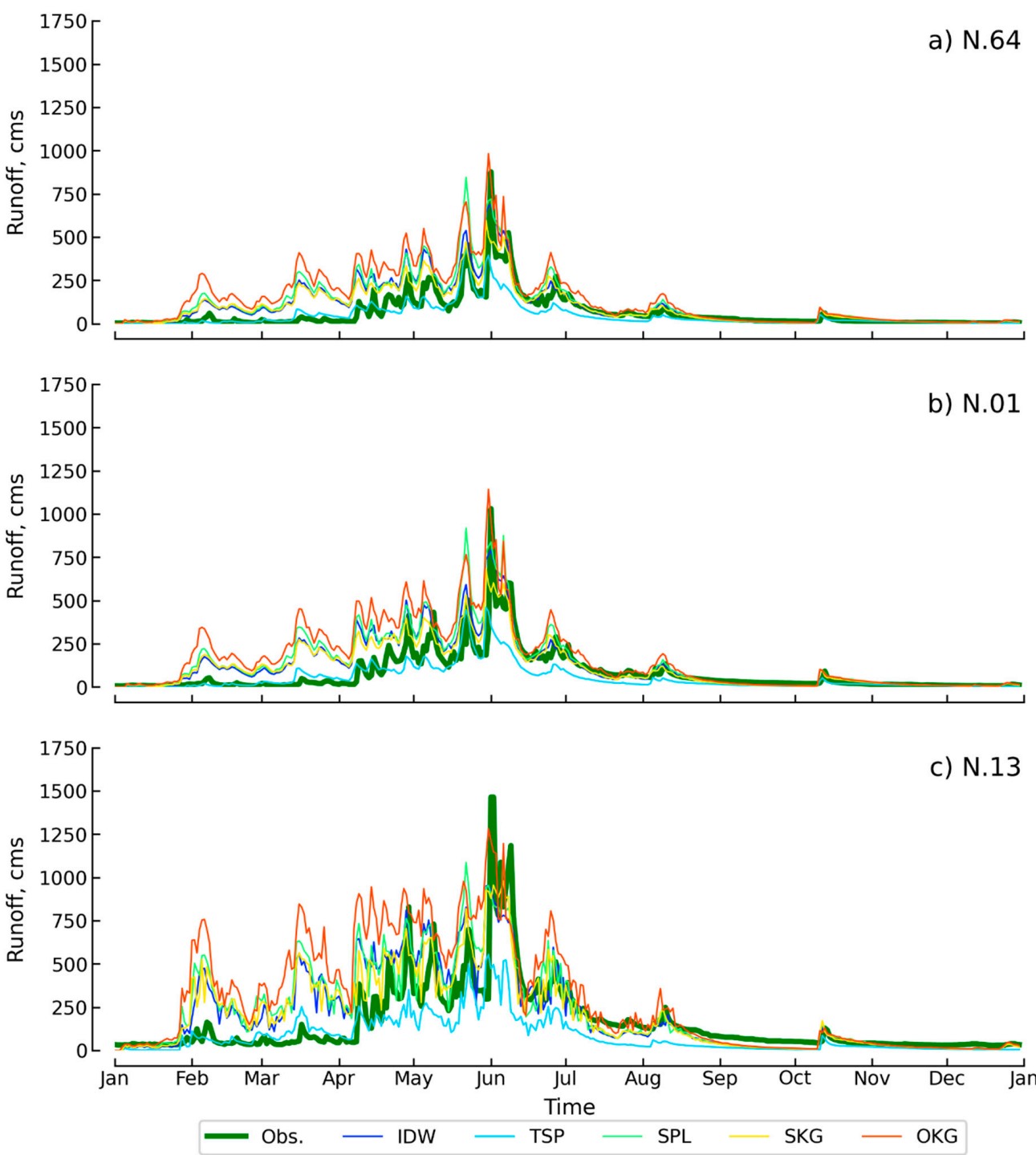

**Figure 8.** Runoff hydrograph at the runoff station for observed runoff and simulated runoff in each rainfall distribution scenario.

**Table 5.** Performance statistics for each rainfall distribution scenario on the runoff simulation.

| Distribution Scenario | Volume Bias, % | Peak Bias, % | RMSE, cms | Correlation | Mean Bias, cms | Nash-Sutcliffe |
|---|---|---|---|---|---|---|
| IDW | 16.88 | −29.20 | 184.12 | 0.755 | 64.19 | 0.437 |
| TSP | −40.59 | −59.46 | 275.05 | 0.691 | −168.02 | 0.427 |
| SPL | 33.25 | −16.73 | 239.53 | 0.704 | 135.89 | 0.223 |
| **SKG** | **6.80** | −35.80 | **164.48** | **0.803** | **27.99** | **0.499** |
| OKG | 43.64 | **1.92** | 275.37 | 0.702 | 185.38 | −0.557 |

Bold is the highlight of the highest accuracy of the distribution scenario.

This study and previous research have mentioned that kriging rainfall distribution significantly impacts runoff simulation accuracy [51]. The spatial distribution of rainfall influences hydrograph shape, peak discharge, and flood volume [16]. Despite the increasing use of kriging in rainfall-runoff modeling, current work on the impact of rainfall spatial distribution [13]. Utilizing the kriging spatially distributed rainfall data improves performance in runoff simulations [52]. In light of the spatial and temporal correlation in basin response, the temporal distribution of rainfall holds greater sensitivity than its spatial distribution, as elucidated in a recent study [53].

## 5. Conclusions

This study emphasizes the significant impact of variable rainfall patterns on estimating river basin discharge during specific storms. The distribution of rainfall within a river basin impacts the water budget and hydrological processes.

To achieve the objectives of this study, the five rainfall interpolation scenarios, IDW, TSP, SPL, SKG, and OKG, were evaluated by using hydrological modeling through the Rainfall-Runoff-Inundation (RRI) model on the Nan River Basin, Thailand. The five interpolation scenarios were evaluated for runoff estimation during the 2014 event without calibration of hydrological parameters. Daily simulated runoffs were compared with measured data from the RID in Thailand.

First, evaluation of the interpolated rainfall results revealed that both kriging scenarios, SKG and OKG, provided the best fit to measured rainfall data, although both scenarios underestimated peak flow with a high ratio of NSE. SPL is an overestimation of runoff volume and mean runoff, while the peak is an underestimation, resulting in low RMSE, high correlation, and low NSE. On the other hand, TSP showed a significant underestimation of the runoff volume, peak flow, and mean runoff. IDW underestimated the peak flow while overestimating runoff volume and mean runoff.

The kriging algorithm based on the semi-variogram model presented the best agreement with measured rainfall among the five spatial interpolation scenarios. SKG emerged as the optimal algorithm for interpolating spatial rainfall data to model streamflow in this study. While OKG also performed well in reproducing observed data, it consistently overestimated the results. SPL and IDW reveal an overestimation of runoff volume and mean runoff, while the peak flow is an underestimation. TSP revealed the most significant underestimation of runoff volume, peak flow, and mean runoff, attended by a low correlation and high RMSE. SKG was applied to the semi-variogram model and stood out as the most suitable interpolation algorithm for runoff estimation at the river basin scale.

This study demonstrates the efficacy of hydrological modeling in revealing the sensitivity of spatial rainfall to river basin responses. The validity of such an analysis, however, is contingent on the faithful reproduction of the watershed response by the model employed. As emphasized in this study, the effect of rainfall variability is dependent on the spatial and temporal characteristics of both the rainfall and the river basin's hydrological attributes. It is worth noting that the spatial and temporal aspects are intricately correlated with rainfall variability in reality. The present study offers valuable insights into spatial daily rainfall variability with a view toward enhancing runoff predictions in hydrological modeling at the river basin watershed scale.

Although this study presents valuable findings, the proposed numerical simulation has several limitations that should be considered. The hydrological modeling was limited to the Nan River Basin in Thailand, and the results may not be generalizable to other regions with different hydroclimatic conditions. Hydrological parameter calibration was not performed, which could affect the accuracy of runoff estimations. The evaluation was based on a specific storm event in 2014, and the results may vary under different climatic conditions or for other storm events. This study relied on traditional interpolation methods, and while the selected kriging algorithm demonstrated effectiveness, there is potential for further improvement through the integration of advanced AI technologies, such as machine learning algorithms. These techniques could enhance rainfall interpolation accuracy, overcome spatial heterogeneity challenges, and improve hydrological modeling predictions by capturing complex data relationships. Therefore, future research should address these limitations and advance the hydrological modeling field.

**Funding:** This research received no external funding.

**Acknowledgments:** This study cannot be conducted without the data provided by the Royal Irrigation Department and the Land Development Department in Thailand.

**Conflicts of Interest:** The author declares no conflict of interest.

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
