# Peer review of "Impact of Spatial Rainfall Scenarios on River Basin Runoff Simulation a Nan River Basin Study Using the Rainfall-Runoff-Inundation Model"

_2673-4117, doi:10.3390/eng5010004_

Round 1

Reviewer 1 Report

Comments and Suggestions for Authors

This study aims to investigate the impact of spatial rainfall distribution scenarios on runoff simulation by using hydrological modeling named Rain- fall-Runoff-Inundation (RRI) model. The topic is interesting and the MS is well written. The reviewer only have some minor comments.

1.      The Nash coefficient (NSE) was recommended to regard as the Performance statistical factors to validate the performance of runoff simulation

2.      Table3, the observed rainfall is recommended to add

3.      Fig5, the scatter plotted was recommended as the column plot was not clear and there are two many colors in the Figure

4.      Fig.8, the runoff simulation based on the OKG performed worse during Mar to Apri, please give some explanations

Comments on the Quality of English Language

The English was accepted

Author Response

Impact of Spatial Rainfall Scenarios on River Basin Runoff Simulation A Nan River Basin Study Using the Rainfall-Runoff-Inundation Model

We would like to thank the editors and reviewers for their comments and suggestions, which improved the quality of the manuscript. Below, we describe how we have addressed these comments and suggestions and the changes made to the manuscript accordingly. The blue text corresponds to our response (R) and paper modifications (PM). PM is reflected in red in the manuscript.

This study aims to investigate the impact of spatial rainfall distribution scenarios on runoff simulation by using hydrological modeling named Rain- fall-Runoff-Inundation (RRI) model. The topic is interesting and the MS is well written. The reviewer only have some minor comments.

1.The Nash coefficient (NSE) was recommended to regard as the Performance statistical factors to validate the performance of runoff simulation

R: The Nash coefficient (NSE) was added to the revised manuscript to present the accuracy of the simulation results from rainfall estimation scenarios. The NSE value for each scenario is also discussed in the revised manuscript.

PM: Page 1, Line 21. Page 8 Line 323-326, Line 335-338. Page 10, Line 413. Tables 4 and 5.

  1. Table3, the observed rainfall is recommended to add

R: The observed rainfall was added to the Table 3.

PM: Table 3. Rainfall Volume for five distribution scenarios over the watershed area.

Rainfall distribution scenario

Rainfall volume, MCM

OBS

7,900.01

IDW

7,045.41

TSP

3,012.01

SPL

7,578.28

SKG

7,787.04

OKG

10,653.01

  1. Fig5, the scatter plotted was recommended as the column plot was not clear and there are two many colors in the Figure

R: The figure was changed to the scatter plot to show the relationship between observed and simulated data for each scenario of rainfall distribution.

PM: Figure 5. Scatter plot of rainfall data for observed and estimated distribution scenarios; a) IDW, b) TSP, c) SPL, d) SKG, and e) OKG.

  1. Fig.8, the runoff simulation based on the OKG performed worse during Mar to Apri, please give some explanations

R: The IDW, SPL, SKG, and OKG reveal the overestimation during March and April due to the period being a dry season, while this study focuses on flood simulation in the wet season. The parameter used to estimate rainfall spatial might be unsuitable for the dry season. Then, using two sets of parameters for dry and wet seasons is recommended. 

PM: Page 9 Line 389-393: Figure 8 presents the simulated daily runoff hydrograph for all five rainfall interpolation scenarios at the measured runoff stations. Among all scenarios during the dry season from January to April, the TSP revealed the simulated value close to the observation data, while other scenarios revealed overestimation. Using two sets of parameters for rainfall estimation specific to dry and wet seasons is recommended.

Reviewer 2 Report

Comments and Suggestions for Authors

The article is titled 'Impact of Spatial Rainfall Scenarios on River Basin Runoff Simulation A Nan River Basin Study Using the Rainfall-Runoff-Inundation Model'. The aim of this paper was to investigate the impact of rainfall spatial distribution scenarios from ground-based observation stations on runoff simulation using hydrological modeling specific to the Rainfall-Runoff-Inundation (RRI) model. The author used the RRI model with six different scenarios of the spatial distribution of input precipitation. The article is interesting, but requires improvement. My comments are as follows:

- The introduction should be supplemented and new literature published after 2020 should be added. The purpose of the work should be improved

- I propose to supplement chapter 2.3 with a short description of rainfall over the last 20-30 years

- Section 4. Presents the results and discussion. The author did not conduct a discussion and did not refer to research by other authors. This part needs improvement.

- Section 5 is a summary, in this part there should be no literature citations, only the main conclusions resulting from the study.

Technical notes:

- the article should be adapted to the editorial requirements of the journal

- The literature is poor, it consists of only 32 items. The author cites only 1 new item, published in 2021. It is necessary to supplement the literature

- fig 5 lacks explanations, what do the individual numbers mean?

​

Author Response

Impact of Spatial Rainfall Scenarios on River Basin Runoff Simulation A Nan River Basin Study Using the Rainfall-Runoff-Inundation Model

I would like to thank the editors and reviewers for their comments and suggestions, which improved the quality of the manuscript. Below, we describe how we have addressed these comments and suggestions and the changes made to the manuscript accordingly. The blue text corresponds to our response (R) and paper modifications (PM). PM is reflected in red in the manuscript.

The article is titled 'Impact of Spatial Rainfall Scenarios on River Basin Runoff Simulation A Nan River Basin Study Using the Rainfall-Runoff-Inundation Model'. The aim of this paper was to investigate the impact of rainfall spatial distribution scenarios from ground-based observation stations on runoff simulation using hydrological modeling specific to the Rainfall-Runoff-Inundation (RRI) model. The author used the RRI model with six different scenarios of the spatial distribution of input precipitation. The article is interesting, but requires improvement. My comments are as follows:

- The introduction should be supplemented and new literature published after 2020 should be added. The purpose of the work should be improved

R: The introduction of the revised manuscript was referred to the new publication journal during the year 2020 as the reference number 3-7, 12-16, 20-24, and 26-29

PM: Page 2, lines 44-54, lines 63-74, and lines 91-97.

- I propose to supplement chapter 2.3 with a short description of rainfall over the last 20-30 years

R: The rainfall description of Northern Thailand, where the study area is located, was explained over 20-30 years.

PM: Page 4 lines 153-163: Over the last 20-30 years, Northern Thailand has seen an increasing trend in precipitation levels [25]. The period of the year characterized by rainfall, typically spanning from June to September, has been observed to commence earlier and extend further, encompassing May to October [26]. In the region, the mean annual rainfall values have been discovered to be most pronounced in the northeastern sector, gradually diminishing towards the southwest [27]. Furthermore, applying clustering techniques to assess the precipitation patterns within this area has also revealed the presence of the highest rainfall zone in the northern section, displaying moderate precipitation levels in the southeastern region and the lowest levels in the southwestern area [28]. Consequently, these findings suggest an overall escalation in precipitation in Northern Thailand over the preceding few decades, alongside variations in rainfall distribution throughout different sections of the region [29].

- Section 4. Presents the results and discussion. The author did not conduct a discussion and did not refer to research by other authors. This part needs improvement.

R: The referred research in the results and discussion section was mentioned for the rainfall spatial distribution scenarios and runoff simulation from the rainfall. 

PM: Page 9 lines 378-381: Among all the spatial interpolation rainfall scenarios, the kriging method with the semi-variogram model offered the best agreement with observed rainfall in this study and consistent with findings from other studies: Tabios and Salas [37], Ly et al. [39], and Ly et al. [44].

Page 9 lines: 415-422: As mentioned in this study and previous research [13, 16, 53, 54], kriging rainfall distribution significantly impacts runoff simulation accuracy. The spatial distribution of rainfall influences hydrograph shape, peak discharge, and flood volume [16]. Despite the increasing use of kriging in rainfall-runoff modeling, current works on the impact of rainfall spatial distribution [13]. Utilizing the kriging spatially distributed rainfall data improves performance in runoff simulations [54]. In light of the spatial and temporal correlation in basin response, the temporal distribution of rainfall holds greater sensitivity than its spatial distribution, as elucidated in a recent study [55].

- Section 5 is a summary, in this part there should be no literature citations, only the main conclusions resulting from the study.

R: The literature citations in the conclusion section was deleted to present only the main conclusions only the results from this study.

PM: Page 10, lines 423-470.

Technical notes:

- the article should be adapted to the editorial requirements of the journal

R: The revised manuscript was improved to follow the format of the journal.

- The literature is poor, it consists of only 32 items. The author cites only 1 new item, published in 2021. It is necessary to supplement the literature

R: The literature was updated up to 55 items, as shown in the reference section of the revised manuscript.

- fig 5 lacks explanations, what do the individual numbers mean?

R: The figure was changed to the scatter plot to show the relationship between observed and simulated data for each scenario of rainfall distribution.

Reviewer 3 Report

Comments and Suggestions for Authors

The manuscript is very well written and ticks most of the important aspects of flood simulation. The numerical modelling is explained in detail and methods are clearly described. Authors may highlight the contribution within the introduction towards the end of section. Further, it would be nice to see limitations of the proposed numerical simulations and how does the numerical modelling can be benifitted using the state of the art AI technologies. 

Author Response

Impact of Spatial Rainfall Scenarios on River Basin Runoff Simulation A Nan River Basin Study Using the Rainfall-Runoff-Inundation Model

I would like to thank the editors and reviewers for their comments and suggestions, which improved the quality of the manuscript. Below, we describe how we have addressed these comments and suggestions and the changes made to the manuscript accordingly. The blue text corresponds to our response (R) and paper modifications (PM). PM is reflected in red in the manuscript.

The manuscript is very well written and ticks most of the important aspects of flood simulation. The numerical modelling is explained in detail and methods are clearly described. Authors may highlight the contribution within the introduction towards the end of section. Further, it would be nice to see limitations of the proposed numerical simulations and how does the numerical modelling can be benifitted using the state of the art AI technologies.

R: The highlight of this study was mentioned in the introduction of the revised manuscript. According to the limitation and future research of this study was recommended in the end of the conclusion section of the revised manuscript.

PM:

The highlight, Page 3, lines 112-117: This study highlights the effectiveness of hydrological modeling in identifying the sensitivity of spatial rainfall on river basin responses. However, the accuracy of such an analysis depends on the precise replication of the watershed response by the employed model. This research offers valuable insights into daily rainfall variability in spatial, aiming to improve runoff predictions in hydrological modeling at the scale of river basin watersheds.

The limitation and future research of this study, Pages 10-11, lines 458-470: Although this study presents valuable findings, the proposed numerical simulation has several limitations that should be considered. The hydrological modeling was limited to the Nan River Basin in Thailand, and the results may not be generalizable to other regions with different hydroclimatic conditions. Hydrological parameter calibration was not performed, which could affect the accuracy of runoff estimations. The evaluation was based on a specific storm event in 2014, and the results may vary under different climatic conditions or for other storm events. The study relied on traditional interpolation methods, and while the selected kriging algorithm demonstrated effectiveness, there is potential for further improvement through the integration of advanced AI technologies, such as machine learning algorithms. These techniques could enhance rainfall interpolation accuracy, overcome spatial heterogeneity challenges, and improve hydrological modeling predictions by capturing complex data relationships. Therefore, future research should address these limitations and advance the hydrological modeling field.

Round 2

Reviewer 2 Report

Comments and Suggestions for Authors

The author has partially improved the article. The discussion is not satisfactory. The inserted literature items should be verified. Some items from the literature list are not included in the text, including: author's items. Moreover, some of the quoted items are repeated too often.

Author Response

Impact of Spatial Rainfall Scenarios on River Basin Runoff Simulation A Nan River Basin Study Using the Rainfall-Runoff-Inundation Model

 I would like to thank the editors and reviewers for their comments and suggestions, which improved the quality of the manuscript. Below, I'd like to describe how we have addressed these comments and suggestions and the changes made to the manuscript. The blue text corresponds to our response (R) and paper modifications (PM). PM is reflected in red in the manuscript.

The author has partially improved the article. The discussion is not satisfactory. The inserted literature items should be verified. Some items from the literature list are not included in the text, including: author's items. Moreover, some of the quoted items are repeated too often.

R: I have thoroughly reviewed and enhanced the manuscript, ensuring that all references are appropriately incorporated into the text. The literature list has been meticulously cross-checked, and any discrepancies have been rectified, including the inclusion of the author's items. Additionally, I have streamlined the reference count from 55 to 53 by excluding my previous journal.

In terms of citation frequency, only seven reference items—namely, [13], [16], [17], [37], [39], [44], and [50]—are cited twice in the manuscript. I believe these revisions address the concerns raised and contribute to an improved overall quality of the article.

PM: Please see the red text as the reference list number.